# PDAC, the Influencer Cancer: Cross-Talk with Tumor Microenvironment and Connected Potential Therapy Strategies

**DOI:** 10.3390/cancers15112923

**Published:** 2023-05-26

**Authors:** Leonardo Mercanti, Maria Sindaco, Mariangela Mazzone, Maria Carmela Di Marcantonio, Mariagrazia Piscione, Raffaella Muraro, Gabriella Mincione

**Affiliations:** 1Department of Innovative Technologies in Medicine & Dentistry, University “G. d’Annunzio” of Chieti–Pescara, 66100 Chieti, Italy; leonardo.mercanti001@studenti.unich.it (L.M.); maria.sindaco@studenti.unich.it (M.S.); mariangela.mazzone@unich.it (M.M.);; 2Campus Bio-Medico University of Rome, 00128 Roma, Italy; mariagrazia.piscione@unicampus.it

**Keywords:** pancreatic ductal adenocarcinoma, PDAC tumor microenvironment, PDAC treatments, PDAC new therapeutic strategies

## Abstract

**Simple Summary:**

The aim of this review was to gather a deeper insight on the mechanisms of Pancreatic Ductal Adenocarcinoma (PDAC), with a particular focus on its biomolecular variety and underlying intracellular and intercellular mechanisms. This was obtained through a critical approach to the current literature on PDAC. We were astounded by the dramatic role played by the Tumoral Microenvironment (TME) in the natural history of this disease, as well as its complexity, which stems from the combination of cellular and acellular components. This, in turn, led us to shift our attention to the current state of PDAC therapy, which at present relies heavily on traditional, invasive techniques. However, recent discoveries such as CAR-T and hyaluronidase-based protocols, give us hope that future approaches will be tailored around each patient’s needs for a better clinical outcome.

**Abstract:**

Pancreatic ductal adenocarcinoma (PDAC) is among the leading causes of death by cancer in the world. What makes this pathological condition particularly lethal is a combination of clinical and molecular heterogeneity, lack of early diagnostic indexes, and underwhelming results from current therapeutic protocols. A major cause of PDAC chemoresistance seems to lie in the ability of cancer cells to spread out and fill the pancreatic parenchyma, exchanging nutrients, substrates, and even genetic material with cells from the surrounding tumor microenvironment (TME). Several components can be found in the TME ultrastructure, including collagen fibers, cancer-associated fibroblasts, macrophages, neutrophils, mast cells, and lymphocytes. Cross-talk between PDAC and TME cells results in the latter being converted into cancer-favoring phenotypes; this behavior could be compared to an influencer guiding followers into supporting his activity. Moreover, TME could be a potential target for some of the newest therapeutic strategies; these include the use of pegvorhyaluronidase-α and CAR-T lymphocytes against HER2, FAP, CEA, MLSN, PSCA, and CD133. Other experimental therapy options are being currently studied, aiming to interfere with the KRAS pathway, DNA-repairing proteins, and apoptosis resistance in PDAC cells. Hopefully these new approaches will grant better clinical outcomes in future patients.

## 1. Introduction

Pancreatic ductal adenocarcinoma (PDAC) is a relatively uncommon cancer, arising from the exocrine pancreas, that is predicted to be the second-leading cause of cancer-related mortality in the United States by 2030 [1,2]. PDAC represents a clinical challenge since 90% of tumors are diagnosed at a late stage, with obvious clinical symptoms, after they have spread beyond the pancreas with systemic metastases (>50%), when surgical resection is no longer feasible, and these tumors are characterized by a peculiar resistance to therapy [3,4].

At present, PDAC is the seventh leading cause of global cancer and has an overall 5-year relative survival rate of approximately 10% in the USA; incidence and mortality (both crude and age-standardized rate) are higher in men than in women and the median advanced age at diagnosis is 70 years [3,5,6].

PDAC arises from non-invasive precancerous lesions, classified as low-grade or high-grade based on the morphological grade of dysplasia of their lining epithelium, curable if detected and treated early enough [2,7]. The most common precursor of invasive PDAC is the pancreatic intraepithelial neoplasms (PanINs), microscopic lesions that occur in the small pancreatic ducts [2,7]. It has been suggested that PanINs may play a role in the development of localized pancreatitis and that the resultant epithelial injury and repair cycles may further propagate the neoplastic process [8]. A smaller proportion of PDACs (<10%) arise from intraductal papillary mucinous neoplasms (IPMNs), macrocystic lesions that involve the pancreatic ductal system and differ from the mucinous cystic neoplasms, the least common, which do not involve the ductal system and have a characteristic ovarian-type stroma [2]. Low-grade PanINs share early somatic changes of Kirsten rat sarcoma virus (*KRAS*) oncogene mutations, while high-grade PanINs are associated with telomere shortening and alteration of the tumor suppressor genes *TP53*, *CDKN2A*, and/or *SMAD4* [2,6,7]. In recent years, surprising advances in sequencing data have demonstrated that normal ductal epithelium, PanINs and PDACs, share similar genetic alterations. In fact, *KRAS* variants are identified in 90% to 92% of patients with PDAC; hence, the possibility of assessing genetic mutations using a non-invasive analysis of human biospecimens is encouraging, both for the early diagnosis of pancreatic cancer (PC) and for the identification of precancerous lesions [1,6,7].

Although the exact etiology of pancreatic cancer remains mostly unknown, advances in understanding potential risk factors have been made in recent years. These identified risk factors can be divided into modifiable and non-modifiable categories [5]. The former includes cigarette and tobacco smoking, excess alcohol consumption, obesity, dietary factors (consumption of red and/or processed meats, sugar-sweetened foods and drinks, foods containing saturated fatty acids and soy products), occupational exposure to toxic substances, chronic pancreatitis, *Helicobacter pylori*, human immunodeficiency virus and hepatitis B/C infection, socioeconomic status and insurance; the latter includes age, gender, ethnicity, AB0 blood group, microbiota (oral, gut, and pancreatic), diabetes mellitus, family history and genetic susceptibility [3,5]. It is well-known that PDAC tumor cells generate their own specific microenvironment and are able to protect themselves from chemotherapy producing an intense stromal reaction [8]. Thus, this intractable malignancy, characterized by invasiveness, rapid progression, and strong resistance to treatment, urgently needs tools for early detection and therapies that can kill cancer cells more effectively after they have metastasized.

What are the future prospects?

Currently, PDAC surveillance is focused on genetically predisposed individuals, since a population-based screening is not currently justified due to its relatively low incidence, compared with other cancers (e.g., breast, colon, or lung cancer). Unfortunately, as mentioned above, most patients develop symptoms, often vague and non-specific, at an advanced stage of the disease, which translates to a delay in diagnosis. Canonical diagnostic tools are still far from being replaced by the use of circulating tumor DNA, a less invasive modality for early detection suffering, however, from low sensitivity and specificity [9].

Surgery, due to advancements in technique, remains the only treatment that offers curative potential in patients where surgical resection is still feasible [9]. The current standard of systemic therapy is represented by FOLFIRINOX (a combination of 5-fluorouracil (5-FU), leucovorin, irinotecan, and oxaliplatin) or gemcitabine plus nanoparticle albumin-bound (nab) paclitaxel, used also for neoadjuvant or adjuvant chemotherapy [2,9]. Chemo and radiation therapies, with their newer delivery modalities, often allow tumors previously designated as inoperable to be operable [9]. To date, other combinations have not shown significant survival benefits over these treatments and/or result in treatment-limiting toxicities [2]. 

The goal of the current research is to translate the cross-talk between tumor cells and the tumor microenvironment into promising therapeutic solutions. The possibility of identifying specific targetable pathways in certain patient subpopulations allows us to personalize therapy and improve their treatment outcome. What emerges from some recent studies is that many subtypes of PDAC susceptible to targeted therapies are associated with specific genetic alterations (e.g., Breast cancer type 1 susceptibility proteins BRCA1 and BRCA2 mutations, microsatellite instability, KRAS mutation, and AT-rich interactive domain-containing protein 1A ARID1A mutation) [2].

Most studies have highlighted the role of pancreatic TME in the progression of PDAC, where different cell types would either restrain the cancer or provide help for invasion and metastasis; the latter category could be a potential target for future therapeutic protocols, some of which are already being tested in trials. The goal of this review is to deepen our understanding of PDAC TME and to summarize and discuss the current state of the art on PDAC therapy options.

## 2. Review Strategies and Literature Included

For this review, the PubMed database was used for the article search. The keywords were “Pancreas adenocarcinoma and microenvironment”, “Pancreas adenocarcinoma and stroma”, “CAR-T and PC”, “Immunotherapy and PC”. For PDAC TME section, papers in the English language that were published (or accepted for publication) between 2017 and 2022 were included. For PC treatment section, papers in the English language that were published (or accepted for publication) between 2014 and 2022 were included. The primary search, after duplicates were removed, provided 1.913 papers. The following important step involved the selection of only the publications in journals with IF > 4 and Q < 2. This led to the inclusion of 188 papers. The following step for the selection excluded all of the papers where data about PDAC treatments were not accurate or not updated and papers where argumentation was off topic (113 removed). A total of 25 relevant articles crucial for the topic were added. Of these, however, 4 do not meet the inclusion criteria, as they have an IF < 4.

After applying these criteria, 100 papers provided the core literature for the current review (Figure 1).

## 3. PDAC TME

The tumor microenvironment (TME) is an altered stroma localized at the interface between the tumor and the healthy parenchyma of the organ [10]. Its presence has been observed in several types of cancer, including PDAC [10]. In this case, TME has been shown to play a pivotal role in tumor development and chemoresistance, to the point that it has been designated as a tumor hallmark [10].

PDAC TME is composed of a dense desmoplastic stroma, in which several cell lines are immersed, including stroma-secerning pancreatic stellate cells (PSCs) and cancer-associated fibroblasts (CAFs), alongside immune cells [10]. Although the characterization of TME and its precise functions is yet to be clarified, there are strong indicators that PDAC’s natural history would be profoundly different in its absence [10]. The significance and heterogeneity of PC stroma have been emphasized by Moffitt et al., who identified two subgroups of PDAC stroma: “normal” and “activated” stroma, with the latter being a malignant, more inflamed version of the former [10].

Moffitt’s [10] findings were further analyzed and compared with other TME studies in a review by Useros et al., which compared information from several major studies [10,11,12,13]. Different terminologies were used in each article, and they needed a side-to-side observation in order to find matches between different classifications [11]. The review lists 4 tumor subtypes (squamous, immunogenic, progenitor, and ADEX), each with its own combination of tumor and stromal class [11].

The squamous subtype is characterized by the highest representation of PSCs and CAFs, along with endothelial cells and TAMs, globally expressing a high number of adhesion molecules (integrins, laminins), growth factors (IGF, VEGF), and inflammation-related genes [11]. These factors contribute to an aggressive phenotype, high chemotherapy (gemcitabine and nab-paclitaxel) and radiotherapy resistance, and reduced T-cell activity inside the specimens [11].

Immunogenic type PDAC comprises a high percentage of immune cells (B and T cells, TAMs) flanking KRAS G12V-positive cancer cells, which also express GATA6 [11]. Overall, the specimens showed resistance to chemotherapy and platinum therapy, tumor immunosuppression, and an augmented expression of immune response-related genes (mostly from the CD and IL families) [11].

Progenitor PDAC is the “simplest” subtype, where the only accessory cell population consisted of type 2 pancreatic ductal cells, overexpressing SOX9 [11]. Tumor cells producing higher quantities of mucin and survival pathways were found to be upregulated, resulting in a poorer clinical prognosis [11].

Finally, ADEX is an endocrine subtype which proved capable of impacting a patient’s hormonal balance; in fact, β-cell destruction is likely caused by the action of endocrine cells and PSCs in the TME [11]. PSCs are also responsible for a general genetic instability and augmented chemoresistance [11]. Although this evidence does not seem encouraging, ADEX cancers showed a better clinical outcome [11].

However, the authors themselves have expressed their doubts on how trustworthy their samples might have been. In fact, mixing human PDAC cells and murine stroma in patient-derived xenografts might have influenced the specimens’ behavior (e.g., desmoplasia and stroma activation). The main components of PDAC TME will be discussed in the following paragraphs.

### 3.1. Acellular Component of PDAC TME

The acellular component of TME consists of an extracellular matrix (ECM), a rigid three-dimensional network of tightly packed proteins and other biomolecules, such as glycosaminoglycans (GAGs) [14]. Most of the protein components of TME (e.g., collagens, GAGs, fibronectin, tenascin) are secreted by PSCs and CAFs, after they are activated by pancreatic cancer cells (PCCs) via the Sonic hedgehog (SHH) pathway [14]. The deposition of ECM components seems to be positively influenced by the tumor itself, in a number of ways. For instance, the expression of missense mutations of *TP53* in PCCs has been associated with increased ECM production by CAFs [15]. Additionally, cancer cells have been reported to produce interleukin-1β (IL-1β), to induce a higher activation of quiescent PSCs, which leads to the synthesis of a greater amount of ECM [15].

Moreover, cancer cells subjected to continuous high-dose chemotherapeutic protocols express higher levels of UDP-N-acetyl-D-galactosamine:polypeptide N-acetylgalactosaminyltransferase-6 (pp-GalNAc-T6) [16]. This enzyme is implied in the glycosylation of Fibronectin, which is converted into oncofetal fibronectin (onf-FN), an ECM component and epithelial–mesenchymal transition (EMT) promoter, exclusive to tumors and embryonic tissues [17]. Increased levels of onf-FN have been observed in cancer cells showing a Multidrug Resistance phenotype, suggesting a role of Fibronectin (and its modifications/interactions with cancer cells) in the increase in chemoresistance [16].

The massive matrix deposition leads to an increased interstitial pressure, which compresses the vessels in the tumor, ultimately causing their collapse [18]. Thus, as cancer progresses, the tissue becomes less vascularized, generating an isolated hypoxic environment, which allows a further evolution of the disease [18]. It also contributes to the phenomena of immuno-escape and chemoresistance by rendering the tumor mass unattainable by blood-mediated immune cells and drugs [18].

The main pressure-enhancing ECM components are GAGs, which expand after binding water molecules [18].

In this hypoxic environment, PCCs are forced to switch to KRAS*-mediated anaerobic metabolism, which results in a massive production of lactate, released in nearby ECM [19]. This metabolite drives the transformation of macrophages into their anti-inflammatory phenotype [19]. Lactate also impairs cytotoxic T cell (CTL) metabolism and hinders their infiltration in the tumor mass [19]. Globally, the result is a weakened immune response towards cancer cells [19].

Regarding the components of ECM, collagens appear to be the most frequent element in the matrix [20]. Type I and type V collagens boost the advancement of the disease, whereas type XV opposes tumor progression [20]. A total of 12% of PDAC cases have highly aligned collagen in the stroma, which is correlated with significantly worse prognosis after tumor surgical resection [20].

### 3.2. PSCs and CAFs

PSCs are star-shaped cells situated mainly around pancreatic acini [14]. They are characterized by a central nucleus surrounded by numerous lipid droplets, which store Vitamin A [14]. They compose about 5–7% of the total pancreatic cells [14]. Given the wide range of biomarkers expressed on their surface and in various cell compartments, the exact origin of PSCs is yet to be defined [17]. Although the mesoderm has been identified as one of the sources, PSCs might also come from neural precursors [17].

PSCs can be activated by pancreatic chronic inflammation, with cytokines and growth factors (e.g., IL-1β, IL-6, tumor necrosis factor alpha, or TNF-α, transforming growth factor beta 1, or TGF-β1) acting as stimulating molecules [21]. These factors drive their differentiation towards a myofibroblast-like phenotype, marked by the expression of CAF-distinctive molecular markers [21]. The main function of activated PSCs is the production of ECM components, contributing to the desmoplastic reaction discussed in the “acellular components” section [14,17].

Stellate cells fuel PCCs by providing substrates for Krebs cycle (e.g., amino acid and palmitate) via exosomes [17]. PSCs scavenge for alanine via autophagy, before releasing it for PDAC cells to absorb and use as an energy source (through its conversion into pyruvate, to feed into the tricarboxylic acid cycle, or TCA) [17]. This allows cancer cells to survive and proliferate in a nutrient-poor environment [17]. PSCs can also help cancer cells via immune inactivation [17]. This can happen through the release of dendritic cell (DC)-suppressing IL-10 and TGF-β1 and the liberation of T cell-inactivating galectin-1 [17]. They also secrete C-X-C motif chemokine ligand (CXCL) 10, which calls for regulatory T lymphocytes (T-regs) and turns off CTL and natural killer (NK) lymphocyte-mediated tumor cell killing [17].

CAFs constitute 15–85% of all the cellular lines present in PDAC TME [20]. Their characterization has been a laborious challenge for researchers, who now seem to agree to opt for different criteria of identification, like molecular markers, cellular shape, and position [20]. The most well-known CAF marker is α-smooth muscle actin (α-SMA) [20]. Yet, it is relevant to notice that activated CAFs also express collagen, type I, alpha 1 (Col1a1), fibroblast activation protein (FAP), fibroblast surface protein (FSP-1), platelet-derived growth factor receptor (PDGFR) β, TGF-β, and podoplanin [20]. CAFs appear as spindle-shaped, elongated, pseudopod-presenting cells [20]. They differ from regular fibroblasts since they cannot be found in healthy tissues, yet they abound in complete or incomplete ring-shaped clusters surrounding the tumor [20]. CAFs seem to be generated from many different cellular predecessors; some of these are even external to the pancreas before their involvement in the PDAC TME, and PSCs give a relatively small contribution to the development of these fibroblasts [22]. In addition to PSCs, CAFs have been shown to develop from epitheliocytes, endotheliocytes, pericytes, adipocytes, quiescent resident fibroblasts, bone marrow (BM)-derived mesenchymal stem cells (MSC), and BM-derived macrophages [22].

In the current review, we decided to focus our attention on the discussion of CAF formation from BM-derived macrophages and MSCs, due to the relevance and actuality of the studies regarding this subject. 

In a study on mouse BM-derived macrophages, Iwamoto et al. demonstrated their ability to transform into CAF-like cells prior to interaction with PDAC cells; when treated with PCC-conditioned media, they started to express CAF-related markers [23]. When they employed the same protocol to human peripheral blood (PB)-derived macrophages, they observed similar outcomes [23].

BM-derived MSCs can serve as a source of CAFs when exposed to extracellular PCC-secreted lactate [23]. It stimulates them to produce α-ketoglutarate (αKG), which then activates Ten-eleven translocation (TET) methylcytosine dioxygenases [23]. This enzyme is responsible for epigenomic reprogramming of MSCs, which results in their differentiation to CAFs [23].

As well as PSCs, the main task performed by CAFs is matrix synthesis; they produce the principal protein components of PDAC stroma, such as collagens, glycosaminoglycans (e.g., hyaluronic acid, chondroitin sulphate), fibronectin, tenascin C, and versican [24].

Since a higher deposition of ECM has been associated with an overall worse outcome, CAF-induced fibrosis has been historically considered as a cancer-favoring mechanism. However, the possibility that it is a mere protective response enacted by CAFs has been highlighted in several past studies [24,25,26]. This hypothesis seems to be supported by the evidence of an even worse prognosis in case of α-SMA+-CAF depletion [14].

The main limit found in these studies resides in the use of murine models, which may show molecular discrepancies and different outcomes when compared to human PDAC specimens. Apart from secreting ECM, CAFs are also able to stiffen it (via lysyl oxidase collagen 1 crosslinking) and to degrade it (through the secretion of metalloproteinases) [20]. By doing this, CAFs contribute to the remodelling of the tumor stroma and enhance interactions between ECM and various cell lines [20].

Indeed, CAFs sustain cancer cells, similar to PSCs, via substrate scavenging and autophagy [22]. ECM secreted by CAFs can be used as an energy source by PCCs, through the digestion of collagen to obtain proline [22].

Furthermore, CAFs can reprogram the immune system in favor of the tumor by secreting macrophage colony stimulating factor (M-CSF) [27]. When absorbed by macrophages, this molecule enhances their production of reactive oxygen species (ROS), which promote their transformation into pro-tumoral M2 macrophages [27].

Finally, CAFs can promote metastasis in various manners, including the enhancement of EMT [20]. For more information on this topic, the readers are advised to consult the “role of TME from precancerous lesions to PDAC” section of the current review.

Even at the end of their life cycle, senescent CAFs are still able to aid the tumor in its progression, both stimulating cancer cells to spread beyond the pancreas (via secreting IL-8) and suppressing immune cells [20]. In other words, they keep an open gateway for prisoners (cancer cells) to escape while keeping the guards (our immune system) distracted.

The remarkable heterogeneity of CAFs was pointed out by Elyada et al., who performed an initial division into two distinct subpopulations, with the possibility of interconversion: myofibroblastic CAFs (myCAFs) and inflammatory CAFs (iCAFs) [28,29]. myCAFs are detectable from the early stages of cancer formation, are localized near the tumor, and express high levels of α-SMA [28,29]. iCAFs seem to appear only once the tumor is fully developed and are confined in a more peripheral position [28,29]. Their low expression of α-SMA is accompanied by the elevated production of proinflammatory cytokines (e.g., IL-6) [30].

The list has been recently extended with the addition of several novel CAF subgroups, including the two discussed below. 

Chen K et al. have recently described a new CAF subgroup: complement-secreting CAF (csCAF) [30]. Identified near PDAC cells during the initial phases of tumor development, they produce complement components, with the potential of modulating the immune TME [30].

Elyada et al. identified antigen-presenting CAF (apCAFs), also located in the adjacency of the tumor [28]. Although their lack of costimulatory molecules suggests their incapability to work as antigen-presenting cells (APCs), they can stimulate cluster of differentiation (CD) 4+ lymphocytes via expression of major histocompatibility complex (MHC) class-II and CD74 invariant chain [28]. Moreover, they have been reported as capable of transmuting into myCAFs [28]. 

### 3.3. Immune Cells 

As previously underlined, the clinical outcome of patients with PDAC is inherently related to the composition of the TME. More specifically, patients with poor prognosis showed signs of a heavier tumor-promoting immune infiltrate (e.g., M0 macrophages, memory B lymphocytes, and neutrophils), whereas cancer-unfavoring immune cells (e.g., CD8+ and CD4+ T cells, naïve B lymphocytes, monocytes, plasma cells, and activated mast cells) were found in specimens from patients with a better outcome [31].

Consistent with the aforementioned observations, some studies explored the relationship between PDAC and its immune microenvironment [19,32,33].

It has been reported that this cancer attempts to modulate the activity of immune cells in the TME so as to induce immune suppression, which in turn favors disease progression [19,32]. This occurs in various manners, a few examples of which will follow [33].

Firstly, PDAC cells produce and release IL-1β, through the toll-like receptor (TLR) 4/NLR family pyrin domain containing 3 (NLRP3) inflammasome signalling axis [32]. IL-1β appears to attract immune cells with tolerogenic functions, such as M2-macrophages, neutrophils, helper T (Th17), regulatory B lymphocytes (B-reg), and myeloid-derived suppressor cells (MDSCs) [32]. Moreover, IL-1β stimulates inactive PCSs to differentiate into desmoplasia-inducing mature fibroblasts; the final product is a stiff matrix, impeding the activity of cytotoxic T cells [32]. 

Secondly, PDAC cells may undergo autophagy in order to downregulate the overall expression of MHC-1 on their surface [19]. This causes the impairment of TNF-dependent signalling in T cells, resulting in their decreased cytotoxic activity [19]. 

In the following sub-paragraphs, the most relevant immune cells found in the PDAC TME will be briefly discussed.

#### 3.3.1. Macrophages

Tumor-associated macrophages (TAM, identified by CD68 positivity) are the most copious immune cell line in the TME of PDAC [10]. From the early stages of PDAC, these cells secrete molecules (e.g., IL-6, IL-10, TGF-β, M-CSF, granulocyte-macrophage colony stimulating factor GM-CSF) that recruit PB-derived monocytes, which will then rapidly differentiate into TAMs [10].

Tumor-associated macrophages present in the context of malignantly inflamed PDAC are mainly polarized towards a tumor-promoting M2 phenotype (CD163+ or CD204+) [10,34]. By contrast, tumor-inhibiting M1 macrophages, expressing human leukocyte antigen-DR isotype (HLA-DR), are prevalent in the pancreatic areas with benign inflammation [10,34]. 

One potential explanation of this evidence was provided by a study by Pratt et al. [34]. They demonstrated that activated CAFs secrete gremlin 1 (Grem1), whose production is gradually enhanced in the progression from PanIN to PDAC [34]. Grem1 inhibits the activity (but not the production) of PCC-produced macrophage migration inhibitory factor (MIF), which would normally induce M1 polarization of TAMs [34]. This results in an enhanced differentiation of the macrophages in the TME towards an M2 phenotype [34]. 

It is important to notice the existence of a study which, in contrast with what was just discussed, localized M2-macrophages further from the tumor mass than M1-macrophages, with their distance from PDAC cells being inversely correlated with the clinical outcome of the patients [35].

A way in which TAMs favor tumour growth is immune suppression. An example of their immune-suppressing activities can be found in their cross-talk with complement components aimed at favoring the escape of PDAC from the complement-mediated cell death [36]. Accordingly, Zhang et al. discovered that TAMs overexpress complement components 1q C1Q A and B [36]. They also prevent CD8+ T cell activation by secreting IL-10 and TGF-β [36].

Another way for these cell lines to help cancer development is the promotion of neovascularization (via the production of angiogenic factors IL-6, vascular endothelial growth factor VEGF, matrix metalloproteinases MMPs) and EMT [36].

Additionally, Iwamoto et al. discovered the capability of BM-derived macrophages to transform into CAF-like cells, prior to cross-talk with PCCS, with the possibility of a subsequent differentiation into CAF subtypes based on the set of cytokines produced by surrounding TME cells [23]. This research team also demonstrated that PB-derived macrophages induce PDAC cells to undergo TME; in a xenograft model co-injected with those macrophages, the downregulation of E-cadherin and collagen IV coexisted with the over-expression of vimentin and fibronectin [23].

#### 3.3.2. Lymphocytes

CD4+ lymphocytes have been found to be abundant in proximity of the tumor [19]. Once recruited to the PDAC TME, CD4+ T lymphocytes are mainly induced to differentiate into anti-inflammatory TH2 cells, which favor tumor development [19,37]. This occurs mainly in two manners. First, TH2 lymphocytes produce IL-4 and IL-13, which maintain PDAC cells by supporting their metabolic reactions [19]. Second, they induce TAM differentiation toward cancer-promoting M2-phenotype [37]. 

Another possible phenotype of CD4+ cells in PDAC TME is TH17 [37,38]. These cells promote PDAC cell mitosis (via cytokines TNF-α and IL-17A), and they enhance fibrosis and angiogenesis [37,38]. They also attract myeloid cells to the TME, which have been proved to induce mutation in the PDAC driver genes, supporting cancer progression [37,38]. TH17 are extremely plastic cells able to differentiate into T-regs [37,38].

T-regs (FOXP3+, CD4+, CD25+) may also start independently invading PC TME from more premature phases of its development [39]. They inactivate CD8+ T cells and promote the differentiation of macrophages and neutrophils into their tumor-promoting phenotypes, M2 and N2, respectively [37]. Hence, they promote tumor progression and immune escape, therefore correlating to a worse prognosis [39].

CD8+ lymphocytes, known for their cell-killing functions, are generally found in remote areas of TME [39]. This is probably because they are attracted to CXCL-12 present in distant PSC-produced stroma [39]. It is important to note, however, that the PDAC cases characterized by higher levels and a more proximal position of CTLs tend to have better clinical outcomes [39].

Once considered as cancer-suppressing cells [40,41], ɣδ T lymphocytes have recently been reconsidered as tumor-favoring [37,42,43]. This happened in light of the fact that these lymphocytes express T-cell exhaustion ligands on their surface: Programmed Cell Death Ligand 1 (PD-L1) and Galectin-9. [37,42].

It is not uncommon to find exhausted T cells in PDAC TME. They have been described as the senescent version of CTLs, resulting from an excessive prolonged activation [39]. These cells are no longer able to perform their physiological cell-killing activity and express a range of inhibitory molecules on their surface [39]. This way, they turn against the immune system by impairing its anti-tumoral action [39].

NKs have been found in diminishing numbers going from a healthy pancreas to a diseased organ, especially in the case of pancreatic tumors expressing mutant KRAS [44]. The prevalence of NKs was considerably lower in areas more densely occupied by PDAC cells than in the surrounding areas of the TME, probably due to the repulsion mechanism of PDAC cells [37,44]. However, some NK populations prevailed in the very proximity of the tumor [44]. Among their functions, NKs are capable of inducing polarization of macrophages in their immune-suppressor M2 phenotype [44].

#### 3.3.3. Neutrophils

Neutrophils are attracted to the PDAC TEM by the tumor cells themselves by secreting chemokines (e.g., CXCL8 and CXCL16) [34]. They are stationed in areas with a higher concentration of cancer cells, increasing in number from the periphery to the center of the tumor [34]. Their current characterization, however, still lacks a clear distinction between the various possible subsets [34].

The ones localized in PDAC TME are mainly tumor-promoting N2 neutrophils that assert their function by secreting different classes of molecules, including pro-inflammatory cytokines (TGF-β and GM-CSF) and ROS [37]. They also show a pro-tumorigenic behavior, which will be discussed in the “metastasis” section [34].

PDAC has demonstrated a high sensitivity to neutrophil activity, especially to lytic enzymes such as arginases-1 [10]. When incubated in an arginase-rich soil, PDAC cells showed augmented levels of Caspase-8 and Bap20 (indicating apoptosis), and cell death through an ER stress pathway was observed [10].

#### 3.3.4. Mast Cells

Mast cells appear as remarkably more abundant in PDAC tissue than in a healthy pancreas [37]. After being engaged to the tumor by PCC-secreted signalling molecules and growth factors (such as VEGF, fibroblast growth factor FGF, RANTES, and C-C Chemokine Ligand CCL2), they in turn act as sentinels, attracting other components of the immune system by chemotaxis [37]. Moreover, they release factors which boost cancer development, angiogenesis, and cell migration (e.g.,VEGF, PDGF, IL-8, secretin, and proteases) [37]. They are also capable of reshaping the tumoral landscape by inducing the expansion of both PCCs and SCs through the secretion of tryptase and IL-13 [17]. All of these actions ultimately drive the patients towards more advanced cancer stages and an earlier exitus [17].

### 3.4. Cross-Talk between PDAC and TME

The importance of the TME is due to its continuous interplay with PDAC cells. This dialogue plays a key role in enhancing tumor development, immune escape, and chemoresistance (Figure 2).

A recent genetic analysis of PDAC samples has led to the discovery of nine fundamental genes and eight interaction pathways that allow cancer cells to dialogue with the stroma surrounding them [45]. These pathways include wound-healing processes and the (re)arrangement of both ECM and actin cytoskeleton [45].

Hedgehog (HH) and TGF-β represent the main factors in the cross-talk between PDAC cells and TME cells [46,47].

HH acts exclusively via paracrine pathways, in the following way: PDAC cells secrete Sonic hedgehog (SHH) and Indian hedgehog (IHH) to aid in the process of tumor progression, according to Dardare J 2020 [37,46,47].

TGF-β is synthesized by T lymphocytes and operates through both autocrine and paracrine pathways, interacting with the numerous target cells exhibiting its receptors [47].

CAFs stimulated by TGF-β acquire antitumoral properties, which would otherwise be restricted [47].

On the other hand, CTLs stimulated by TGF-β show a pro-carcinogenic behavior, stimulating progress from PanIN to PDAC (which in turn appears to slow down upon removal of the above-mentioned molecules) [46].

It has been observed that TGF-β plays a dual role in pancreatic epithelial cells during cancer progression, where different pathways prevail in different stages of the disease [46]. In healthy cells, TGF-β binds to TGF-βR-I and II, activating receptor-regulated SMAD (R-SMAD) proteins. R-SMADs bind to co-mediator SMADs (co-SMAD), such as SMAD4, which bind to the SMAD Binding Element (SBE) in the nucleus. If SMAD4 is wild-type, as it happens in healthy pancreatic cells, it will initiate a Sox4-mediated apoptotic process [46]. TGF-β also operates through SMAD-independent pathways (MAPK, PI3K) to promote cell growth and proliferation [46]. During earlier stages of PDAC, TGF-β acts as a tumor suppressor by using the SMAD-dependent (canonical) pathway, whereas during later stages, it likely relies on the non-canonical pathway [46]. Switching from canonical to non-canonical pathways, TGF-β favors tumor progression in two ways. First, it enhances the expression of EMT factors Snail and Zeb1/2 [46]. Second, as previously mentioned, TGF-β-stimulated NKs and T Lymphocytes are silenced while T-reg populations proliferate [46]. Late-stage tumors also show alterations determining the inactivation of SMAD4, which could play a role in TGF-β switching from the canonical to the non-canonical pathway [46,47].

Besides the HH pathway, tumor cells may engage in interactions with CAFs in another manner.

A study on a murine model carried out by Djurec et al. highlighted Serum Amyloid A3 (Saa3), belonging to the family of serum amyloid A apolipoproteins, as a key molecule involved in the cross-talk between PDAC cells and CAFs [48]. To prove this, while the Saa3-expressing CAFs are known to promote tumor growth, the ones which do not secrete Saa3 showed tumor-restraining activities [48]. This may be due to the latter subgroup overexpressing membrane palmitoylated protein 6 (Mpp6), a member of the MAGUK family [48]. Mpp6 can induce the formation of multiprotein complexes that suppress cancer expansion [48]. It is relevant that Saa3 is no more than a pseudogene in the human genome; hence, it is not found in human models of PDAC cell lines [48]. However, the acute-phase protein Saa1 is thought to be a human ortholog protein of Saa3, which could play a similar role in humans [48].

It is important to notice that cross-talk is not exclusive to PDAC cells interacting with TME cells, as it is also largely used by cell populations within TME in order to communicate to each other. 

Specifically, CD11c-expressing DCs interact with thymic Naive T-regs shortly after their arrival in the microenvironment, resulting in the latter differentiating in CD44+ memory cells [49]. The result of this pas de deux is the inactivation of a DC’s capability of alerting our immune system, which in turn reduces CTL intervention in the tumor [39]. Thus, the TME reinforces the tumor by silencing the immune sentinels (APCs) and preventing any external attacks [49]. 

Cells could also engage in the aforementioned pathways through extracellular vesicles (EVs), which come in three different variants, based on their dimensions: exosomes or small EVs (40–200 nm), ectosomes or micro vesicles (50–2000 nm), and apoptotic bodies (500–4000 nm) [50]. Out of the three subgroups, exosomes appear to be the smallest EV type as well as the most frequently used means of molecular exchange among TME cells and PDAC cells [50]. They are small vesicles composed of a lipid bilayer shell containing several classes of biomolecules, including nucleic acids, glycans, proteins, and lipids [50]. Their production among TME and PDAC cells occurs through multivesicular bodies (MVBs), which merge with the cell membrane to release them [50]. Then, they diffuse in the extracellular space, until they reach the target cells, by which they will be absorbed, altering their metabolism [50]. Small EVs are thought to be involved in cancer progression, immune escape, and metastasis [50]. 

The exosome communicating system has been reported in several cell lines [50,51,52,53,54]. Due to their large number, we chose to exclusively select the most relevant ones for the present review.

M2 macrophages appear to boost tumor growth by releasing EVs packed with microRNA-301a-3p, related to hypoxic conditions and capable of downregulating TGF-β receptor (TGFβR) 3, enhancing TGF-β activation [51]. EVs are received by PDAC cells, which gain a more aggressive and invasive phenotype, seemingly favoring lymph nodes and vases as their go-to option. In addition, EVs push PDAC towards EMT [51]. This is proven by the fact that EV-stimulated PDAC cells downregulate the expression of E-cadherin, while upregulating the expression of molecules involved in migration and angiogenesis [51]. 

One way for CAFs to aid tumor growth is the secretion of a set of EVs which contain five different miRNAs (21a, 92a, 181a, 221, 222) [52]. When cancer cells uptake these vesicles, several pathways are altered, as a result of the miRNAs’ interaction with the cellular genome [52]. One of these is the phosphoinositide 3-kinase (PI3K)/protein kinase B(AKT) pathway, which includes phosphatase and tensin homolog (PTEN), a tumor-suppressing gene [52]. Hence, by impairing PTEN gene expression through the inhibition of its upstream pathway, miRNAs accelerate tumor progression [52]. 

Additionally, they secrete exosomes to induce the creation of lysyl-oxidase-mediated crosslinks in the ECM; this action renders it thrice stiffer than normal, thus promoting chemoresistance [55]. The precise mechanism by which this event occurs is yet to be completely understood, but recent studies hypothesized that the modulation of the yes-associated protein (YAP) pathway may be involved [55].

Moreover, CAFs secrete EVs containing molecules (lactate, acetate, Krebs cycle intermediates, lipids, amino acids) which can be used as fuel by PDAC cells prior to a switch from aerobic to anaerobic metabolism [53]. CAFs that have just undergone autophagy secrete peculiar exosomes of this kind [53] that contain alanine, which may provide nourishment for PCCs in a context of glucose paucity [53]. 

PDAC cells secrete small biomolecule-containing EVs via the fusion of MVBs with the cell membrane [50]. These EVs are carried to adjoining and distant healthy parenchymal cells that may include them by endocytosis, direct membrane fusion, or receptor–ligand bond [50]. This results in the impairment of a healthy cell physiological function, driving it towards neoplastic transformation and favoring cancer expansion [50].

Furthermore, tumor-derived exosomes (TEXs) may contain molecules involved in the induction of a tolerogenic TME, by initiating CD8+ apoptosis, repressing NKs and stimulating T-regs and myeloid cells [54].

### 3.5. The Role of TME from Precancerous Lesions to PDAC

Tumor microenvironment plays a crucial role in all phases of PDAC natural history, modulating its own features and functions to better cooperate with the tumor itself to endorse cancer formation, invasion, and metastasis (Figure 3). Several changes, both in molecular expression and cellular infiltrates, were reported in the TME during the evolution from precancerous lesions into proper PC.

Studies conducted on PanIN have brought some interesting results to our attention [56,57].

Progression towards PDAC is marked by an increased presence of immune-suppressing cells in the TME, such as T-regs and M2 macrophages, other than the expansion of PSCs [56]. Although its general levels appear to be influenced by diet, fibronectin also appears to be augmented in the process [56].

It seems to us that the role of fibroblasts in the progression from pre-cancer lesion to the tumor itself is worthy of particular attention. Hence, it will be briefly discussed as follows.

Favored by a chronically inflamed pancreas (e.g., chronic pancreatitis), the epitheliocytes of the acini undergo acinar-ductal metaplasia (ADM) [56]. Simultaneously, these degenerate cells, which express mutated KRAS and myelocytomatosis oncogene (MYC), stimulating the differentiation of PSCs in activated fibroblasts [46]. Activated CAFs lose their cytoplasmic lipid droplets and start to express α-SMA and to secrete ECM components (collagen, fibronectin, laminin, and hyaluronic acid), as well as inflammatory cytokines and growth factors [56].

In this early stage of the tumor, CAFs directly boost cancer development by releasing TGF-β1, which prolongs the expression of oncogenic MYC by new PDAC cells [57]. 

Once the in situ PDAC is formed, it quickly develops an invasive front along the edge of the tumor [56]. CAFs, found exclusively in the invasive front (while the non-invasive one is occupied only by cancer cells and fibrosis), promote desmoplasia and tumor growth, at the expense of nearby pancreatic acini that proceed towards atrophy [56].

In the meantime, cancer cells induce angiogenesis by liberating angiogenic factors (e.g., VEGF-A, TNF-α), allowing interactions with surrounding cells (e.g., CAFs, pericytes, and endothelial cells) [57]. These cells are recruited and activated by the tumor so that they can act as the proper builders of the newly formed tumoral vascular system [57]. Cross-talk between the endothelial cells of the newly formed vessels and cancer-inducing cells (CICs) supports the latter’s maintenance and growth, favoring cancer expansion [57].

TME elements also play a crucial role when PDAC starts to invade distant tissues, favoring metastasis.

In particular, it seems that blood-travelling PCCs are accompanied by CAFs, which enhance their mobility and protect them [20]. CAFs also contribute to the formation of the ideal metastatic niche for tumor cells to comfortably grow in it, as proven by their premature presence in liver metastasis composed only by 6–7 cells [20].

Interestingly, as soon as they reach the liver, PCCs secrete integrin-expressing exosomes which stimulate hepatic TCs to start deposing ECM [20]. The newly produced matrix will quickly reconstitute PDAC TME for the hepatic metastases as well [20].

Neutrophils and neutrophil extracellular traps (NETs) are also able to enhance the metastatic process [35]. These cellular elements loosen both ECM and endothelial junctions, boosting EMT and allowing circulating cancer cells to infiltrate distant organs, such as the liver [35]. Here, NETs also provide a sort of safety net for circulating elements, which can bind more easily to liver sinusoids [35].

## 4. Conventional Treatment in Resectable Pancreatic Cancers

PDAC has the highest percentage of death among solid cancers. Despite the introduction of new therapeutic strategies, the prognosis is terribly poor since metastasis renders this cancer inoperable, and only 20% of all patients with PDAC are eligible for surgery. Thus, to increase the chances for successful treatment, it is important to focus on detecting symptomatic patients as early as possible.

An ideal screening test must be efficient, reliable, and safe for the patient, with the highest sensitivity and specificity. Traditional imaging technologies, such as CT scans and MRI, are not suitable for initial screening; they can only be used if the patient is symptomatic [58]. There are no specific PDAC-related symptoms; therefore, early detection seems very difficult if not impossible [58].

Nevertheless, there are five biomarkers (ApoA1, CA125, CA19-9, CEA, ApoA2, and TTR) which are suitable for PDAC early diagnosis [58]. 

At the moment, CA19-9 is the most effective biomarker, with an 80% specificity and a 79% sensitivity [58]. However, this biomarker is more effective in the follow up of PDAC treatment rather than in its diagnosis [58]. It is for this reason that the therapeutic approach to the pathology is of great interest [58].

Localized pancreatic cancers may be divided into resectable, borderline resectable, and locally advanced (the latter infiltrates superior mesenteric vessels).

If the tumor is resectable, adjuvant treatment consists of 5-fluorouracil, leucovorin, irinotecan, and oxaliplatin (FOLFIRINOX) [59]. This standard of care ensures the longest median overall survival (54 months) [59]. By contrast, the combination of nab-paclitaxel plus gemcitabine or FOLFIRINOX has a little impact on overall survival [59,60].

However, the results of two trials, ESPAC-4 and PRODIGE-24, propose different treatment options for resectable PDAC [59]. A total of 732 patients with resectable PDAC were randomized to receive either adjuvant gemcitabine plus capecitabine or single-agent gemcitabine [61]. The addition of capecitabine extends the median overall survival by 3 months [61].

New therapeutic strategies are affected by the fact that PDAC has recently been defined as a systemic disease (Figure 4).

Sohal et al. discussed preclinical and clinical data and argued that even early-stage PDAC does not consist of a local disease [62]. Indeed, autopsy demonstrated that 70–85% of patients with early-stage PDAC died of systemic recurrence and not of local disease, after PC resection [62].

In a randomized phase II/III Prep-02/JSAP-05 trial, 364 Japanese patients with resectable PDAC were randomized and divided into two groups [63]. Patients from the first group underwent upfront surgery followed by adjuvant combination of tegafur, gimeracil, and oteracil (S-1) for 6 months [63]. The second group received neoadjuvant gemcitabine plus S-1 as a first step, followed by resection and adjuvant S-1 [63]. 

The results highlighted an increased rate of R0 resections (no cancer cells were present microscopically at the primary tumor site) in patients who were treated with neoadjuvant therapy, despite pharmacokinetic differences between ethnicities [64].

To summarize, at present, patients with a good performance status had a positive outcome after mFOLFIRINOX treatment. Adjuvant chemotherapy for 6 months with mFOLFIRINOX represents the current standard for treatment.

Considering the possibility of a relapse, a popular randomized trial compared mFOLFIRINOX with gemcitabine. A total of 247 people were assigned to receive mFOLFIRINOX. A total of 80 of them discontinued treatment. Only 15 had a relapse, corresponding to 6% [59].

At the same time, attention must be brought to those patients who cannot receive mFOLFIRINOX, gemcitabine plus capecitabine treatment. In patients with borderline resectable disease, neoadjuvant treatment is necessary for downstaging and R0 resections [65].

According to Napoli3 (a randomized, open-label, phase III clinical trial for metastatic PDAC), first-line NALIRIFOX (liposomal irinotecan administered with 5-fluorouracil/leucovorin) improved the overall survival and the progression-free survival compared with Gemcitabine + NabPaclitaxel in treatment-naïve patients with metastatic PDAC [66]. Moreover, NALIRIFOX must be considered safe and manageable, Wainberg, Z.A, 2023 [66].

Moreover, chemoradiation (chemotherapy combined with radiotherapy) has long been used in locally advanced PC. Its use, however, has been greatly questioned in the LAP07 study [67]. It has been shown that, after four months of gemcitabine, stable patients achieved similar results whether they continued gemcitabine therapy or switched to chemoradiation (54Gy with capecitabine) [67]. 

Regarding other forms of radiation therapy, newer studies are taking radiofrequency ablation, irreversible electroporation, high-intensity focused ultrasound, and microwave ablation into consideration [67]. 

In conclusion, since PDAC is a systemic neoplasm from an early stage, the success of any local approach, other than surgical resection and adjuvant therapy, is very limited [67]. 

## 5. New Therapeutical Strategies against PDCA

### 5.1. Pathway Inhibition

RAS (Rat sarcoma) genes, HRAS (Harvey RAS), KRAS (Kirsten RAS), and NRAS (Neuroblastoma RAS) represent the most frequently mutated oncogene family in human cancer [68]. Indeed, mutations on these genes have been demonstrated in three of the most lethal cancers of the United States (lung, colorectal, and pancreatic cancer) with an interest percentage of about 25–30% [68]. 

It is important to state that PanIN contains RAS mutations, increasing the possibility of cancer development [68,69]. However, despite the strenuous effort to find an anti-RAS therapy, no effective RAS inhibitors have been shown and there is not a single effective RAS inhibitor for all RAS mutated cancers [68,69]. However, new strategies have been implemented since KRAS mutations are detected in 95% of PDAC [68,69].

A novel inhibitor of *KRAS* G12C (ARS-1620) has been shown to inhibit tumor growth in in vivo preclinical models [70]. Like other KRAS G12C inhibitors, such as (AMG510), it seems to be safe and well-tolerated [71]. Nine patients who did not suffer from PDAC were treated with this inhibitor [71]. Six of them had stable diseases while a partial response was detected in one patient [71]. However, this specific kind of mutation accounts for only ~1% of all *KRAS* mutations observed in PDAC [68]. 

At present, the use of RNAs targeting *KRAS* G12D, a strategy which involves exosomes and small EVs as carriers, represents a novel approach, together with combination therapy with novel molecules. Nonetheless, the targeting of *KRAS* still remains difficult to implement [72,73]. Two different research groups have demonstrated that the inhibition of the MAPK signalling pathway with an ERK or MEK inhibitor in *KRAS*-mutant PDAC cell lines elicits an increase in autophagy [72,73]. Pharmacological inhibition of the phosphatidylinositol-3 kinase (PI3K) pathway by using AKT inhibitors with single-agent strategies has produced negative results both in vitro and in vivo [72,73]. However, several completed or ongoing clinical trials have evaluated or are evaluating combinations of inhibitors of specific components of the rapidly accelerated fibrosarcoma (RAF) and PI3K effector pathways [72,73].

### 5.2. DNA Repair

Poly (ADP-ribose) polymerase (PARP) enzymes are involved in DNA damage repair (DDR). They bind to single-strand DNA breaks and recruit other DDR proteins. To act, PARP needs to be released from the DNA helix, so that the replication of fork61 is allowed. 

Cancer cells with mutations that inhibit damage repair via other pathways are often sensitive to PARP inhibitors [65]. 

Kaufman et al. first studied PARP inhibition with olaparib in patients with germline *BRCA1/2* mutation and recurrent cancers [74]. This phase II study was addressed to patients who had failed chemotherapy and who suffered from ovarian, breast, pancreatic, and prostate cancer [74]. 

PARP inhibitors (including olaparib, talazoparib, and rucaparib) differ in their ability to trap PARP. 

For example, in a phase II trial, olaparib, which has a greater trapping ability than veraparib, was investigated as a monotherapy in patients with advanced-stage malignancies and a germline BRCA 1/2 mutation. It has to be taken into account that this trial included 23 patients with PDAC. This trial revealed a 21.7% response rate in this subgroup, with a complete response in one patient and a partial response in four patients [74]. 

What is more interesting are the results of the POLO trial. It is the first randomized trial involving patients with PDAC [75]. Patients with germline BRCA mutations and stable disease after 4 months of platinum-based therapy were randomized to receive olaparib or placebo as maintenance therapy. Olaparib is very well-tolerated (dose intensity 99%). The study met its primary progression-free survival endpoint (median 7.4 months versus 3.8 months; HR 0.53; *p* < 0.004) [75].

Moreover, patients with “BRCAness” represent an important area of research. BRCAness includes tumors with the same molecular characteristics of BRCA-mutant tumors which may positively respond to similar therapeutic approaches. The concept of ‘BRCAness’ refers to cancers with a defect in homologous recombination repair, mimicking BRCA1/2 loss. This kind of neoplasm is sensitive to platinum-based treatment or PARP inhibitors [76]. Recent studies have evaluated the role of PARP inhibitors in patients with PDAC who may not have a germline BRCA1/2 mutation, but share ‘BRCAness’. In particular, a phase II study, which considered patients with ‘BRCAness’, demonstrated that 10–20% of PDAC patients have a DDR deficiency without BRCA mutations and olaparib shows an antitumor activity in platinum-sensitive, BRCA-negative PDAC [75].

### 5.3. Immunotherapy

As discussed in the “PDAC TME” section, hypoxia and fibrosis protect the tumor mass from the patient’s immune system. Thus, strengthening the immune response could be an effective therapeutic approach. Moreover, immunotherapy could help replenish a patient’s immune cell pool after cytotoxic therapies.

It is important to optimize timing, dosage, and choice of therapies [77]. Pembrolizumab, an anti-PD-1 (Programmed cell death protein 1) antibody, was initially approved by the Food and Drug Administration for a tissue-agnostic indication in patients with MMR-deficient malignancies [78]. Pembrolizumab strengthens the immune response against cancer cells by stimulating PD-1 in CTLs [79].

At present, the guidelines recommend MMR genetic tests in patients with advanced neoplasms [80]. However, the presence of a desmoplastic stroma is to blame for the poor results [81]. In spite of that, it has been reported that patients with desmoplastic melanoma had a high response to anti-PD-1 or anti-PD-L1 antibodies (overall response rate 70%) [81]. Binnewies et al. tried to justify the lack of activity of Pembrolizumab in patients with PDAC with the presence of an immunosuppressive TME [82]. A case series report showing a potential role of Pembrolizumab in patients with PDAC at different stages was published in 2022 [83]. The main limitation of this work was the lack of a control group or blinding, so that bias could not be removed [83].

#### Extracellular Tumor Microenvironment

Early studies tend to demonstrate a therapeutic potential in stroma-modulating strategies [84]. The most advanced stromal modulator, the pegvorhyaluronidase-α (PEGPH20), disassembles stromal proteins, increases intratumoral blood flow, and improves progression-free survival in a phase II trial, when added to chemotherapy [84]. However, treatment with this molecule was unsuccessful in phase III clinical trials [84]. 

Moreover, it has been proved that PEGPH20 may be useful for radio-sensitization of PDAC if the TME shows an accumulation of hyaluronan [84].

### 5.4. CAR-T and CAR-M Cell Therapy

Immunotherapy has recently been added to the clinicians’ arsenal in the fight against PDAC. Starting only with CAR-T approaches, physicians are now able to apply CAR-M protocols as well (Figure 5). 

#### 5.4.1. First Use of CAR T Cell Therapy 

CAR-T cell therapy was first used in patients with hematologic malignancies. Kochenderfer et al. showed a partial response to anti-CD19 CAR-T cell therapy in a patient with treatment-refractory stage IVB follicular lymphoma [72]. Unfortunately, after only 8 months, the patient relapsed [85]. 

Prospective studies, containing this and other trials, reported positive response rates ranging from 52 to 92% [86]. 

Moreover, the phase II ELIANA trial of CTL019 CAR-T cell therapy reported the following outcomes in pediatric and young adult patients with relapsed and refractory acute lymphoblastic leukemia: a 75% relapse-free survival probability 6 months after remission; an 89% probability of survival at 6 months; a 79% chance of survival at 12 months. [87]. 

#### 5.4.2. CAR-T Cell Therapy in Solid Tumors: Pancreatic Cancer

Therapeutic protocols based on the use of CAR-T are now used in preclinical models of PC [88]. 

Currently used antigens include: Mesothelin (MSLN), Carcinoembryonic Antigen (CEA), Human Epidermal Growth Factor Receptor 2 (HER2), Fibroblasts Activating Protein (FAP), and Prostatic Stem Cell Antigen (PSCA) [88]. 

What is challenging about targeting solid tumor antigens is that they are also expressed in normal tissues. Therefore, a strenuous selection is necessary not only to reach therapeutic efficacy but also to limit off-tumor adverse events.

Further development of antigen-selecting strategies is necessary in order to obtain antitumor efficacy and minimize toxicity [89]. In this way, CAR-T cell protocols will be suitable not only for hematological malignancies but also for solid tumors [89].

The glycoprotein Msln is a tumor differentiation antigen [90]. It is physiologically found on mesothelial cells in the pleura, peritoneum, and pericardium [90]. It is overexpressed by a variety of neoplasms, including PC [90]. 

In PDAC, Msln activates the NF-kB (nuclear factor kappa- light-chain-enhancer of activated B cells) pathway and determines cell proliferation, by autocrine or paracrine IL-6 stimulation [90]. 

Msln-expressing neoplastic cells upregulate antiapoptotic proteins (such as “B cell lymphoma-XL” and MCL-1) through the Akt/NFκB/IL-6 pathway, resulting in apoptosis inhibition [90]. 

In addition, recombinant endotoxin and vaccines targeting MSLN effectively prevent the proliferation, invasion, and metastasis of PCCs in vivo and in vitro [90]. More than 10 clinical trials using CAR-T cells directed against MSLN have been completed [90] or are still in progress. Six patients with chemo-resistant PDAC were involved in a phase I clinical trial [90]. After CAR-T treatment (three times a week for three weeks), two patients achieved progression-free survival of 3.8 and 5.4 months [90]. Metabolic activity in tumors, studied through metabolic imaging, remained stable in three patients, while it decreased by 68.3% in another patient whose liver metastases completely disappeared [90]. In addition, none of the patients showed signs of CAR-T-related side effects (e.g., CRS) nor dose-related toxicity [90]. 

Moreover, Beatty et al. discussed two case reports using Mesothelin-directed CAR-T cells (CARTmeso cells) [91]. The first patient had a malignant pleural mesothelioma, while the second one was diagnosed with PC [91].

The major goal of the study was to ensure CAR-T safety, exploiting target antigens on normal tissues [91]. For this reason, the authors developed a strategy for transient CAR expression, via mRNA electroporation encoding for an anti-mesothelin ss1 scFv CAR [91]. 

The CARTmeso cells were administered in eight doses of intravenous infusion [91]. In addition, the patient with PC had two intratumoral injections [91]. 

CARTmeso transgene was found in peripheral blood in both patients, in addition to being found in the ascitic fluid 3 days after the initial infusion [91]. CARTmeso transcripts were also found in the tumoral bioptic tissue before and after the surgery in the patient who received the first intra-tumoral injection of CARTmeso cells [91]. These findings indicate that CARTmeso cell trafficking into the tumor occurs after intravenous administration [91]. Finally, it was observed that the patient with pleural mesothelioma developed a partial response, which lasted 6 months, while the patient with PC achieved a stable outcome [91]. 

The same research group conducted a more recent phase I study on CARTmeso cells. Six patients with treatment-refractory metastatic PDAC were administered CARTmeso cells intravenously three times per week for three weeks. This trial reported dose- limiting toxicity, CRS and neurological complications in the patients [92]. However, stable disease was reported in two of the six subjects [92].

Another potential target, Prominin 1 (CD133), is highly expressed in PDAC stem cells, as well as in other neoplasms [92]. 

A phase I trial included seven patients with advanced PC with tumors showing 50% or greater CD133 expression [92]. Treatment led to three patients achieving stable disease and two partial remissions, while the remaining two patients showed progression of their disease [92]. It should be noted that CD133 did not appear in biopsies after treatment, suggesting that any CD133+ cells were eradicated [92].

Meanwhile, other targeting antigens in preclinical models of PDAC continued to emerge. 

Among these, there are interesting data about CAR-T cells against CEA (CEACAM5), an antigen which can be useful for immune response in advanced gastrointestinal malignancies [93]. CEA is considered a target for vaccine trials [93]. 

In a murine model of PDAC, CAR T cells directed against CEA produced long-term anti-tumor responses, with no evidence of damage to normal tissues with lower levels of CEA expression [94]. CEA is highly expressed in approximately 65–75% of pancreatic cancers [90]. Moreover, knockout of the CEA family gene CEA-related cell adhesion molecule (CEACAM) is able to significantly decrease the proliferation of cancer cells in vitro and to increase the total survival time of mice bearing PDAC in vivo [90]. 

Katz et al. conducted a phase I clinical trial on CAR-T therapy, targeting CEA-positive liver metastases from malignant tumors [95]. Out of the six patients involved in the trial, one survived for 23 months with stable disease and no serious CAR-T-related adverse events occurred [95]. These remarkable results may suggest that such a therapy may be applied in patients with high tumor burden who are not responsive to conventional therapy [95]. 

One major challenge for CAR-T cell therapy in preclinical models of PDAC is the strong desmoplastic reaction in many specimens [96]. Defeating this immunosuppressive microenvironment is actually a strategy to improve CAR-T cell survival and numerous preclinical studies are underway [97].

For example, heparanase is under investigation as a means to overcome the desmoplastic reaction [97]. Moreover, heparanase was shown to increase antitumor activity and tumor infiltration of CAR-redirected T lymphocytes in preclinical studies [97].

Furthermore, to address inefficient trafficking of CAR-T cells into the tumor microenvironment, some studies have attempted to target surface molecules related to PDAC [98]. For example, FAP has been considered as a potential target antigen in the context of CAR-T cells [98]. This antigen is expressed on myofibroblast cells present within the pancreatic stroma [98].

Tran et al. developed a novel CAR-T cell approach in which cells were redirected to interact with FAP and transferred into mouse models [98]. However, this strategy was associated with cachexia and lethal bone toxicity, which most likely would limit its application as a universal target with CAR-T cell therapy [98].

Regarding HER2, another clinical study, still in phase I, was drawn up: an active immunotherapy study with a combination of two chimeric (Trastuzumab-like and Pertuzumab-like) HER-2 B Cell Peptide vaccines, emulsified in ISA 720 and the Nor-MDP (muramyl dipeptide) adjuvant [99]. The purpose of this phase I trial, which is currently enrolling into the extension arm, is to evaluate the side effects and best dose of vaccine therapy in treating patients with metastatic solid tumors [99].

#### 5.4.3. Side Effects of CAR-T Cell Therapy

CAR-T therapy may lead to cardiac and systemic toxicity. 

An example of CAR-T-mediated toxicity is Cytokine Release Syndrome (CRS), characterized by aspecific symptoms such as fever, myalgia, fatigue, and mild hypotension. CRS is caused by elevated plasmatic levels of inflammatory cytokines (cytokine storm), which leads to multiple-organ failure, if left untreated. The main pathogenic factors are IL-1, IL-6, and TNF-α, with a correlation between their levels and severity of the disease. 

Many predictive scales have been elaborated to evaluate the intensity of the reaction, the time-to-fever interval (the interval between the administration of the therapy and the onset of fever), and the peak temperature.

Screening patients with sepsis is also crucial, since an immunosuppressive therapy, while trying to resolve CRS, could do more harm than good.

#### 5.4.4. CAR-M Cell Therapy as Another Chance towards Solid Malignancies

Even though CAR-T cell therapy is undeniably effective, at present, chimeric antigen receptor macrophages (CAR-M) therapy should be also considered.

Until November 2020, the Food and Drug Administration approved two clinical trials based on the use of CAR-M cells on various solid tumors [100]. 

The first trial treated relapsed/refractory tumors overexpressing *HER2* with anti-HER2 CAR-M cells [100]. The second one used anti-Mesothelin CAR-M [100]. 

What is promising about CAR-M therapy is that it presents many advantages when compared to CAR-T. Firstly, macrophages can significantly immerse in the tumor environment. Secondly, CAR-M can reduce the ratio of tumor-associated macrophages, with a positive impact on cancer treatment [100]. Thirdly, CAR-M cells spend a limited time in blood circulation and are less cytotoxic for non-tumor tissue [100].

## 6. Conclusions

So far, PDAC has proved itself to be a far more complex clinical entity than previously thought; its innate ability to infiltrate a healthy parenchyma, disrupt the host’s immune system, and spread throughout the organism makes it a particularly difficult target for current therapy protocols. 

The cancer’s crucial strategies reside in its matrix secretion abilities (via PSC activation and cross-talk with CAFs), alongside a complex network of interactions with immune cells, both innate (macrophages, neutrophiles, and mast cells) and adaptive (lymphocytes); the combination of all these factors has turned the research on PDAC into a medical trench war, where every bit of progress comes at an overwhelming price.

It is, however, interesting to note how such a “chaotic” and “degenerate” cluster of cells can show such elaborate interactions with the surrounding environment. 

One of the main challenges is finding an “organic” approach to this disease, with the use of monoclonal antibodies or CAR-T protocols. Although still at earlier stages of development, these therapeutic approaches might pave the way for better, more refined techniques of targeting PDAC in the future. 

## Figures and Tables

**Figure 1 cancers-15-02923-f001:**
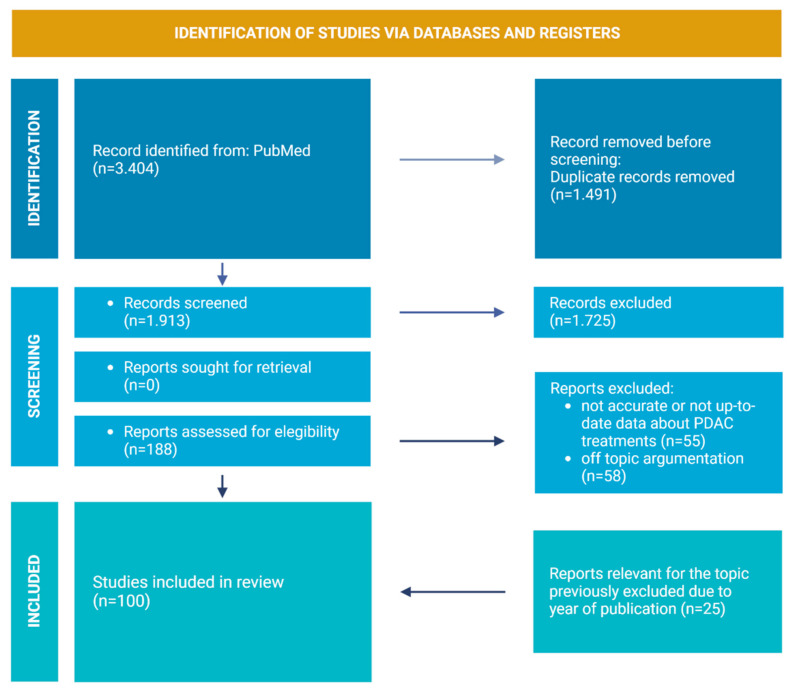
Flow chart of literature selection. Created with BioRender.com.

**Figure 2 cancers-15-02923-f002:**
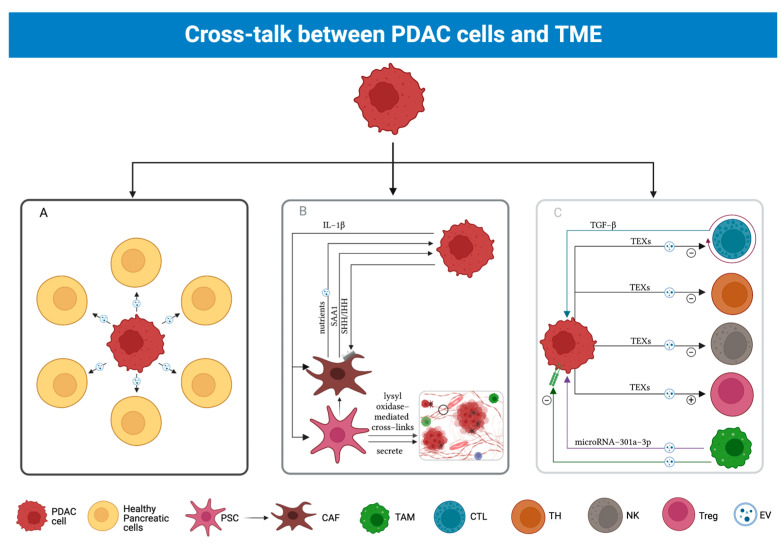
Cross-talk between PDAC cells and TME. PDAC cells have been shown to interact with other cells in the TME. Healthy cells (**A**) are influenced by PDAC towards carcinogenesis; stellate cells and fibroblasts (**B**) are redirected towards an ECM-deposing phenotype through the action of SHH/HH and SAA1 pathways, while also providing PDAC cells with nutrients; immune cells (**C**) receive tumoral exosomes (TEXs) containing miRNAs which will stimulate pro-carcinogenic and pro-metastatic pathways. PSC: pancreas stellate cell; CAF: cancer-associated fibroblast; TAM: tumor-associated macrophage; CTL: cytotoxic T lymphocyte; TH: helper T lymphocyte; NK: natural killer lymphocyte; T-reg: regulatory lymphocyte; EV: extracellular vesicle; Black circle in panel B: cross-link. Created with BioRender.com.

**Figure 3 cancers-15-02923-f003:**
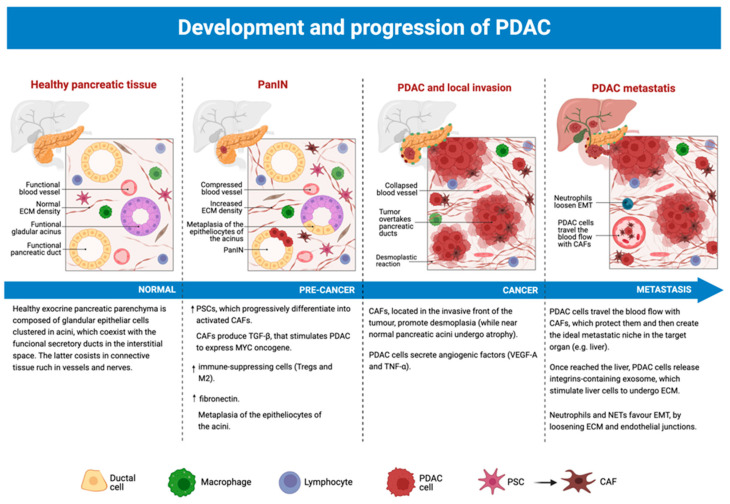
Development and progression of PDAC. Visualization of TME role during the natural evolution of PDAC. PSC: pancreas stellate cell; CAF: cancer-associated fibroblast; ECM: extracellular matrix; PanIN: pancreatic intraepithelial neoplasia; EMT: epithelial–mesenchymal transition; arrow represents an increase. Created with BioRender.com.

**Figure 4 cancers-15-02923-f004:**
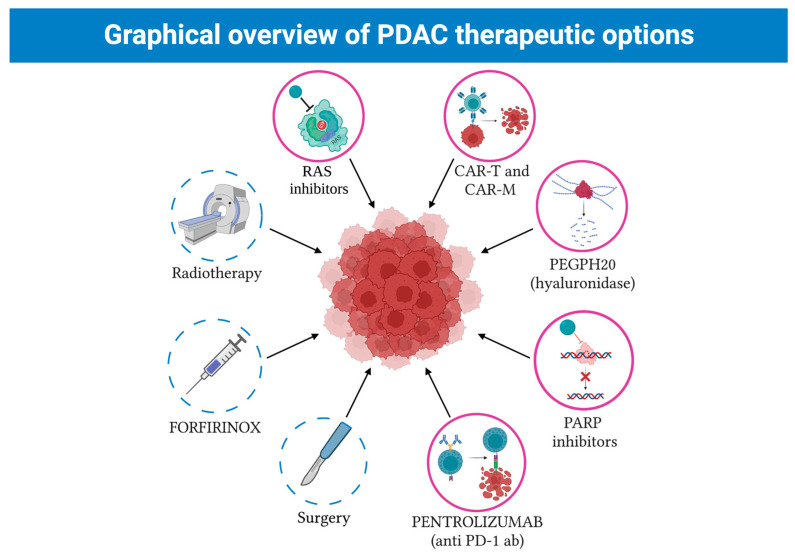
Graphical overview of PDAC therapeutic options. The image above shows currently used protocols (dotted blue line) alongside newer, experimental methods (full purple line). Created with BioRender.com.

**Figure 5 cancers-15-02923-f005:**
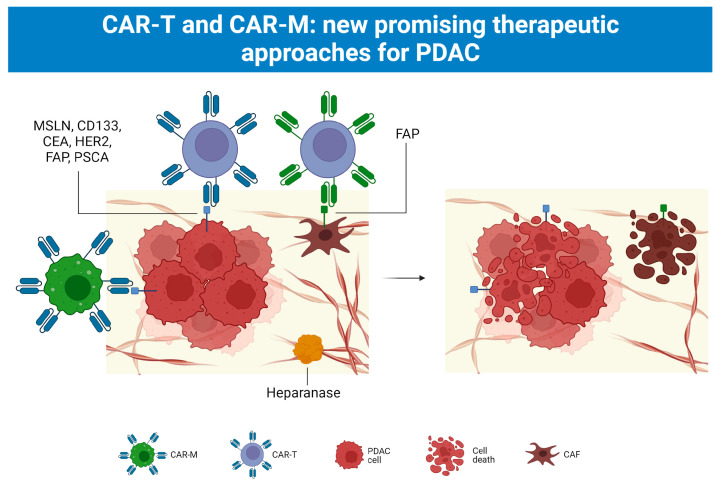
CAR-T and CAR-M are among the latest most relevant therapeutic approaches for PDAC. Once the engineered T lymphocytes or macrophages interact with the target cells (in this case PCCs and CAFs), they induce their death in various ways. MSLN: mesothelin; CD133: promin 1; CEA: carcinoembryonic antigen; HER2: human epidermal growth factor 2; FAP: fibroblast activation protein; PSCA: prostate stem cell antigen. Created with BioRender.com.

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
