# Peer review of "PDAC, the Influencer Cancer: Cross-Talk with Tumor Microenvironment and Connected Potential Therapy Strategies"

_cancers, 2023, doi:10.3390/cancers15112923_

Round 1

Reviewer 1 Report

Summary:

In this review article Mercanti et al., explore about the understanding of PDAC tumor microenvironment, and discuss the current state of the art on PDAC therapy options. I have some suggestions to improve the quality of the manuscript. Please, find my suggestions below.

Major Concern

- The authors could further enrich the topic that talks about PDAC and TME (topic #3). It is already well described that cells such as cancer-associated fibroblasts, endothelial cells, and immune cells comprise the tumor microenvironment of pancreatic cancer. Furthermore, pancreatic cancer is considered an immune‐quiescent disease, but activation of immunological response in pancreatic cancer may contribute to favorable outcomes. Could the authors discuss about it in this topic?

- In topic 3.1, the authors discuss chemoresistance, and the influence of acellular components in this process. However, very little is discussed about which of these components play a role in the acquisition and /or maintenance of chemoresistance. For the topic to be more attractive and appealing to readers, the authors need insert the following references [PMID: 36882122, PMID : 24452447, PMID: 32908154, PMID: 1738241, PMID: 26592953], and make a discussion (even briefly) about this subject within the topic. There are studies showing that tumor cells that express ECM components or/are cultured in the presence of pre-treated plates with laminin, collagen or fibronectin, become resistant to cytotoxic insults mediated by different classes of chemotherapeutic agents.

- Lines 424-426 "When stimulated by TGF-β, PDAC cells will finally undergo apoptosis after a brief EMT phase if they express wild-type SMAD4, whereas they will replicate favouring tumour progression if they express mutated SMAD". Could the authors discuss more in depth on this subject?

- Lines 557-558 "If the tumor is resectable, adjuvant treatment consists of 5-fluorouracil, leucovorin, irinotecan and oxaliplatin (FOLFIRINOX)". Although this is the best therapeutic strategy (if the tumor is resectable) what is the probability of relapse after this type of intervention? Are there papers that bring this type of information?

- In topic 5 4. CAR-T Cell Therapy, could the authors (even briefly) also introduce the importance of CAR-M Cell Therapy? Is this technology already well described for PADC? This is a hot topic in cancer research. I understand that because it is a new technology, the literature is still a little scarce. However, as it is a current issue, and of great interest in the field of oncobiology, if possible, the authors could talk how CAR-M has helped to establish the viability of this innovative immunotherapy, which advances the trailblazing scientific discovery of CAR T cell therapy? 

- In topic 5 5. The authors list a series of promising inhibitors to combat PADC. However, little is said about the selectivity of these drugs. The authors could argue about this subject. Certainly, this kind of discussion will enrich the topic.

Final Comments

Overal the manuscript is well written and has scientific merit. However, I will recommend for publication after the modifications suggested above.

Author Response

Summary:

In this review article Mercanti et al., explore about the understanding of PDAC tumor microenvironment, and discuss the current state of the art on PDAC therapy options. I have some suggestions to improve the quality of the manuscript. Please, find my suggestions below.

Major Concern

Q1. The authors could further enrich the topic that talks about PDAC and TME [topic #3]. It is already well described that cells such as cancer-associated fibroblasts, endothelial cells, and immune cells comprise the tumor microenvironment of pancreatic cancer. Furthermore, pancreatic cancer is considered an immunoquiescent disease, but activation of immunological response in pancreatic cancer may contribute to favorable outcomes. Could the authors discuss about it in this topic?

A1. We appreciate very much for the constructive suggestions. We have added a short discussion of this topic in the section 5.3: “The inaccessible TME is characterized by hypoxia and fibrosis which isolate the mass rendering the tumour immunologically “cold”. Thus, it is important to search for immunotherapies which strengthen local and systemic immune responses. Moreover, therapies which determine the death of immunogenic cells, such as chemotherapy and ablative treatment, may be combined with immunotherapy. Thus, it is important to optimize timing, dosage, and choice of therapies [77]”.

Q2. In topic 3.1, the authors discuss chemoresistance, and the influence of acellular components in this process. However, very little is discussed about which of these components play a role in the acquisition and /or maintenance of chemoresistance. For the topic to be more attractive and appealing to readers, the authors need insert the following references [PMID: 36882122, PMID: 24452447, PMID: 32908154, PMID: 1738241, PMID: 26592953], and make a discussion [even briefly] about this subject within the topic. There are studies showing that tumor cells that express ECM components or/are cultured in the presence of pre-treated plates with laminin, collagen or fibronectin, become resistant to cytotoxic insults mediated by different classes of chemotherapeutic agents.

A2. We thank the reviewer for suggesting these articles. We found “PMID 36882122” to be especially appropriate for our review, and decided to cite it in section 3.1: “Moreover, cancer cells subjected to continuous high-dose chemotherapeutic protocols express higher levels of UDP-N-acetyl-D-galactosamine:polypeptide N-acetylgalactosaminyltransferase-6 [pp-GalNAc-T6] [Reis JSD, 2023]. This enzyme is implied in the glycosylation of Fibronectin, which is converted into oncofetal fibronectin [onf-FN], an ECM component and epithelial-mesenchymal transition [EMT] promoter, exclusive to tumours and embryonic tissues [Reis JSD, 2023]. Increased levels of onf-FN have been observed in cancer cells showing a Multidrug Resistance phenotype, suggesting a role of Fibronectin [and its modifications/interactions with cancer cells] in the rising of chemoresistance [16]”.

Q3. Lines 424-426 "When stimulated by TGF-β, PDAC cells will finally undergo apoptosis after a brief EMT phase if they express wild-type SMAD4, whereas they will replicate favouring tumour progression if they express mutated SMAD". Could the authors discuss more in depth on this subject?

A3. We thank the reviewer to bring these reported researches to our attention. The following is an amplified version which goes more in depth about the role of TGF-β and SMAD proteins in PDAC (section 3.4). “It has been observed that TGF-β plays a dual role in pancreatic epithelial cells during cancer progression, where different pathways prevail in different stages of the disease [46]. In healthy cells, TGF-β binds to TGF-βR-I and II, activating receptor-regulated SMAD [R-SMAD] proteins. R-SMADs bind to co-mediator SMADs [co-SMAD], like SMAD4, which bind to the SMAD Binding Element [SBE] in the nucleus. If SMAD4 is wild-type, as it happens in healthy pancreatic cells, it will initiate a Sox4-mediated apoptotic process [46]. TGF-β also operates through SMAD-independent pathways [MAPK, PI3K] to promote cell growth and proliferation [46]. During earlier stages of PDAC, TGF-β acts as a tumour suppressor by using the SMAD-dependent [canonical] pathway, whereas during later stages it likely relies on the non-canonical pathway [46]. Switching from canonical to non-canonical pathways, TGF-β favours tumour progression in two ways. First, it enhances the expression of EMT factors Snail and Zeb1/2 [46]. Second, as previously mentioned, TGF-β-stimulated NKs and T Lymphocytes are silenced while T-reg populations proliferate [46]. Late-stage tumours also show alterations determining the inactivation of SMAD4, which could play a role in TGF-β switching from the canonical to the non-canonical pathway [37,46].”

Q4. Lines 557-558 "If the tumor is resectable, adjuvant treatment consists of 5-fluorouracil, leucovorin, irinotecan and oxaliplatin [FOLFIRINOX]". Although this is the best therapeutic strategy [if the tumor is resectable] what is the probability of relapse after this type of intervention? Are there papers that bring this type of information?

A4. We added in the text the following sentences: “Adjuvant chemotherapy for 6 months with modified FOLFIRINOX is currently the reference. Taking into account the possibility to have a relapse, in one of the most popular randomized trial, the modified FOLFIRINOX was compared with gemcitabine. 247 people were assigned to receive modified FOLFIRINOX. 80 of them discontinued treatment. Only 15 have a relapse, corresponding to 6% [59]” (section 4).

Q5. In topic 5.4. CAR-T Cell Therapy, could the authors [even briefly] also introduce the importance of CAR-M Cell Therapy? Is this technology already well described for PADC? This is a hot topic in cancer research. I understand that because it is a new technology, the literature is still a little scarce. However, as it is a current issue, and of great interest in the field of oncobiology, if possible, the authors could talk how CAR-M has helped to establish the viability of this innovative immunotherapy, which advances the trailblazing scientific discovery of CAR T cell therapy?

A5. Following the interesting reviewer suggestion, we have added a short paragraph on this “hot topic” in the section 5.4 named: CAR-M cell therapy as another chase towards solid malignancies”.

Q6. In topic 5 5. The authors list a series of promising inhibitors to combat PADC. However, little is said about the selectivity of these drugs. The authors could argue about this subject. Certainly, this kind of discussion will enrich the topic.

A6. We agree and we have added the following in the section 5.4.2: “What is challenging about targeting solid tumor antigens is that they are also expressed in normal tissues. Therefore, a strenuous selection is necessary not only to reach therapeutic efficacy but also to limit off-tumor toxicity. However, development of further strategies to select antigens will be necessary so that antitumor efficacy will be reached and toxicity will be minimized and CAR-T cell therapies will be used not only in hematological malignancies but also in solid tumors [90]”.

Reviewer 2 Report

These authors try to review the possible development of therapy for PDAC. The title is not clear and a bit of syntax that needs to be corrected.

Good 

Author Response

These authors try to review the possible development of therapy for PDAC. The title is not clear and a bit of syntax that needs to be corrected.

We would like to thank you for your suggestions.

Following your input, we have revised our review in order to make ore accessible to international readers by simplifying the syntax and eventually correcting mistakes.

However, we were not able to find an alternate, clearer title, while keeping the idea of an “influencer” disease: the way we see it, this metaphor is appropriate for communicating the concept of someone/something relying on the compliance of surrounding elements in order to survive and thrive in an otherwise hostile environment.

Reviewer 3 Report

This review provides an overview of pancreatic ductal adenocarcinoma (PDAC), covering clinical challenges, the tumor microenvironment, genetic alterations, risk factors, current treatment options, and future prospects. The clinical and molecular heterogeneity of PDAC, the lack of early diagnostic markers, and the limited success of current therapeutic approaches are highlighted. The role of the tumor microenvironment components, such as collagen fibers, cancer-associated fibroblasts, immune cells, and genetic material exchange, in shaping a cancer-favoring environment is discussed. Potential therapeutic strategies targeting the tumor microenvironment, including pegvorhyaluronidase-α, CAR-T lymphocytes, and inhibitors of key signaling pathways and apoptosis resistance, are also explored.

However, there are areas that could be further improved to enhance the quality and impact of the review:

1.     Readability and organization can be enhanced by incorporating clearer paragraph breaks and subheadings.

2.     Language can be simplified, and technical terms and abbreviations can be defined or explained to improve accessibility for readers.

3.     Specific references can be included for each statement or claim to allow readers to access the original sources.

4.     Critical analysis and evaluation of the evidence can be provided, including discussing limitations of the studies mentioned and presenting conflicting viewpoints.

5.     The content can be condensed and focused on the most relevant and significant points to avoid repetition.

6.     The significance and relevance of the "Review strategies and literature included" section and the inclusion of a figure (Figure 1) outlining the methodology of the literature search in a scientific review can be reconsidered.

7.     The last sentence of the manuscript reads “This section is mandatory, with one or two paragraphs to end the main text”. Is it something from the “instructions for authors”?

Addressing these refinements will enhance the overall quality and impact of the review article

Author Response

This review provides an overview of pancreatic ductal adenocarcinoma [PDAC], covering clinical challenges, the tumor microenvironment, genetic alterations, risk factors, current treatment options, and future prospects. The clinical and molecular heterogeneity of PDAC, the lack of early diagnostic markers, and the limited success of current therapeutic approaches are highlighted. The role of the tumor microenvironment components, such as collagen fibers, cancer-associated fibroblasts, immune cells, and genetic material exchange, in shaping a cancer-favoring environment is discussed. Potential therapeutic strategies targeting the tumor microenvironment, including pegvorhyaluronidase-α, CAR-T lymphocytes, and inhibitors of key signaling pathways and apoptosis resistance, are also explored.

However, there are areas that could be further improved to enhance the quality and impact of the review:

Q1. Readability and organization can be enhanced by incorporating clearer paragraph breaks and subheadings.

A1. We thank the reviewer for pointing this out. As suggested, we have revised the general disposition of paragraphs. The paragraphs in the article are now smaller, and more frequent breaks have been made. We’ve also added a new paragraph in the section 5.4 named: CAR-M cell therapy as another chase towards solid malignancies”.

Q2. Language can be simplified, and technical terms and abbreviations can be defined or explained to improve accessibility for readers.

A2. We appreciate very much for the constructive suggestions. We have provided to make the manuscript more accessible for readers.

Q3. Specific references can be included for each statement or claim to allow readers to access the original sources.

A3. We followed the suggestion of reviewer and we have included the references for each statement and claim.

Q4. Critical analysis and evaluation of the evidence can be provided, including discussing limitations of the studies mentioned and presenting conflicting viewpoints.

A4. We appreciate very much for the constructive suggestion; we included a critical analysis of the evidence in the section 3.2 and 5.4.2.

Q5. The content can be condensed and focused on the most relevant and significant points to avoid repetition.

A5. We followed the suggestion of reviewer and we condensed and focused on the most relevant and significant points.

Q6. The significance and relevance of the "Review strategies and literature included" section and the inclusion of a figure [Figure 1] outlining the methodology of the literature search in a scientific review can be reconsidered.

A6. We agree with the referee that a narrative review is less methodologically demanding than a systematic review, as it does not require a search of all literature in a field. However, we think that the search strategy summary of a narrative review makes more transparent the reporting.

Q7. The last sentence of the manuscript reads “This section is mandatory, with one or two paragraphs to end the main text”. Is it something from the “instructions for authors”?

A7. We apologize for any confusion. We have removed the sentence from the manuscript.

Reviewer 4 Report

The article entitled: “PDAC, the influencer cancer: cross talk with tumour microenvi- 2 roment and connected potential therapy strategies” by Mercanti et al., is well written, structured and introduced. Although the article is well written, I recommend the a review for English style, grammar and usage. Please, find some points to improve the quality of the manuscript:

1)The article must follow the International Consensus for genes (upper case and italics), Proteins (upper case and regular) and mRNA (lower case).

2)Authors state that beverages containing high amount of fructose is a risk factor. Please verify if the real risk factor is fructose or glucose and include a reference.

3) Please include a specific reference of the risk factor “soy product” and specify which gender is associated to higher risk of PDAC.

4) Please replace the word “create” by other words like “originate” “generate” etc., since only God can create.

5) Include which is/are the current FDA approved biomarker/s for PDAC.

6) Include which is the best adjuvant treatment according to Napoli-3 clinical trial for metastatic PDAC.

7) In Section 2. Verify that you have selected articles IF>4 and specially "Q<2".

8) In Section 3. It would be interesting to include a match between the molecular subtypes and their tumor microenvironment. Authors can use: Cancers (Basel). 2021 Jan 17;13(2):322. doi: 10.3390/cancers13020322.

9) Section 3.3.3. It should describe the role of arginase released by neutrophils. Authors can use: Sci Rep. 2021 Jun 15;11(1):12574. doi: 10.1038/s41598-021-91947-0)

10) In section 5.4. I miss another figure explaining CAR-T in PDAC.

Some sentences must be review and the use of some words like "possible".

Author Response

The article entitled: “PDAC, the influencer cancer: cross talk with tumour microenvironment and connected potential therapy strategies” by Mercanti et al., is well written, structured, and introduced. Although the article is well written, I recommend the review for English style, grammar, and usage. Please, find some points to improve the quality of the manuscript:

We would like to thank the reviewer for his thoughtful review of the manuscript and we appreciate very much for the positive comments.

Q1. The article must follow the International Consensus for genes [upper case and italics], Proteins [upper case and regular] and mRNA [lower case].

A1. We thank the reviewer for noticing it. We have carefully revised the manuscript and we corrected it according to referee’s indications.

Q2. Authors state that beverages containing high amount of fructose is a risk factor. Please verify if the real risk factor is fructose or glucose and include a reference.

A2. We overlooked this information in the former version of the manuscript, and we thank the reviewer for noticing it. Cai et al. in their review “Advances in the epidemiology of pancreatic cancer: Trends, risk factors, screening, and prognosis”, well describe the risk factors for pancreatic cancer. The table 1 of Cai et al. review summarizes the factors and the risk associated. Sugar-sweetened foods and drinks are associated with moderate risk increase. The high-fructose corn syrup is the major source of caloric sweeteners in soft drink, especially in the United States. However, we agree with the reviewer that this part of the discussion was not particularly clear, and consequently we have decided to replace “foods and beverages containing fructose” with “sugar-sweetened foods and drinks”.

Q3. Please include a specific reference of the risk factor “soy product” and specify which gender is associated to higher risk of PDAC.

A3. As mentioned above, Cai et al. in their review reported a large prospective cohort study conducted in Japan that showed that higher intake of soy products, particularly non fermented soy food, might increase the risk of pancreatic cancer, with an RR of 1.48 (95% CI, 1.15–1.92), published by “Yamagiwa Y, Sawada N, Shimazu T, Yamaji T, Goto A, Takachi R, Ishihara J, Iwasaki M, Inoue M, Tsugane S; JPHC Study Group. Soy Food Intake and Pancreatic Cancer Risk: The Japan Public Health Center-based Prospective Study. Cancer Epidemiol Biomarkers Prev. 2020 Jun;29(6):1214-1221. doi: 10.1158/1055-9965.EPI-19-1254. Epub 2020 Mar 13. PMID: 32169996”. According to the results obtained by Yamajiwa Y et al. no gender association was observed.

Q4. Please replace the word “create” by other words like “originate” “generate” etc., since only God can create.

A4. We appreciate very much for the constructive suggestion. We replaced the word “create” with “generate”. Regarding the citation “Created with BioRender.com”, BioRender Publication Guide recommended to accompany all completed graphics by this citation (for more details it is possible to consult the “Confirmation of Publication and Licensing Rights” of figures in the Annex.

Q5. Include which is/are the current FDA approved biomarker/s for PDAC.

A5. We thank the reviewer to bring this point to our attention. “PDAC has the highest percentage of death. Despite the introduction of new therapeutic strategies, the prognosis is terribly poor since, at the diagnosis, it is inoperable, if metastatic and only 20% of all patients with PDAC are eligible for surgery. What is important to focus on is an earlier diagnosis. An ideal screening test must be efficient, reliable, safe for the patient with the highest sensitivity and specificity. Adopting imaging technologies such as computed tomography or magnetic resonance imaging is not proper for initial screening and they can be used if the patient is symptomatic. There are no PDAC- related specific symptoms; therefore, early detection seems very difficult if not impossible. Nevertheless, there are five biomarker (ApoA1, CA125, CA19-9, CEA, ApoA2, and TTR) which are suitable for PDAC early diagnosis. At the moment CA19-9 is the most effective biomarker since its specificity is of 80% and its diagnostic sensitivity of 79%.  Actually though, this biomarker results very effective to detect the recurrence or the response to a particular treatment instead of to firstly detect PDAC. It is for this reason that the therapeutic approach to the pathology is of great interest [58]”. We included this new paragraph in section 4.

Q6. Include which is the best adjuvant treatment according to Napoli-3 clinical trial for metastatic PDAC.

A6. We appreciate very much for the constructive suggestion and we included following in section 4: “According to Napoli3, a randomized open-label, phase 3 clinical trial for metastatic PDAC, first-line NALIRIFOX (liposomal irinotecan administered with 5-fluorouracil/leucovorin) improves the overall survival and the progression free survival compared with Gemcitabine+NabPaclitaxel in treatment-naïve patients with metastatic PDAC. Moreover, NALIRIFOX has to be considered safe and manageable [66]”.

Q7. In Section 2. Verify that you have selected articles IF>4 and specially "Q<2".

A7.  We thank the referee for the suggestion. We checked again the references and we modified the number of the articles included since we have added the references suggested by the others referees. 4 relevant articles crucial for the topic were added however, do not meet the inclusion criteria as they have an IF < 4.

Q8. In Section 3. It would be interesting to include a match between the molecular subtypes and their tumor microenvironment. Authors can use: Cancers [Basel]. 2021 Jan 17;13[2]:322. doi: 10.3390/cancers13020322 (Martinez-Useros J et al., 2021).

A8. We thank the reviewer to bring this reported research to our attention. Section 3 has been expanded citing the suggested article as below: “Moffitt’s findings were further analysed and compared with other TME studies in a review by Useros et al., which crossed information from several major studies [10,11,12,13]. Different terminologies were used in each article, and they needed a side-to-side observation in order to find matches between different classifications [11]. The review lists 4 tumor subtypes (squamous, immunogenic, progenitor, and ADEX), each with its own combination of tumour and stromal class [11].

The squamous subtype is characterised by the highest representation of PSCs and CAFs, along with endothelial cells and TAMs, globally expressing a high number of adhesion molecules (integrins, laminins), growth factors (IGF, VEGF) and inflammation related genes. [11]. These factors contribute to an aggressive phenotype, high chemotherapy (gemcitabine and nab-paclitaxel) and radiotherapy resistance, and reduced T-cell activity inside the specimens [11].

Immunogenic type PDAC comprises a high percentage of immune cells (B and T cells, TAMs) flanking KRAS G12V positive cancer cells, which also express GATA6 [11]. Overall, the specimens showed resistance chemotherapy and platinum therapy, tumour immunosuppression, and an augmented expression of immune response related genes (mostly form the CD and IL families) [11].

Progenitor PDAC is the “simplest” subtype, where the only accessory cell population consisted of type 2 pancreatic ductal cells, overexpressing SOX9 [11]. Tumour cells produced higher quantities of mucin and survival pathways were found to be upregulated, resulting in a poorer clinical prognosis [11].

Finally, ADEX is an endocrine subtype which proved capable of impacting a patient’s hormonal balance: in fact, β-cell destruction is likely caused by the action of endocrine cells and PSCs in the TME [11]. PSCs are also responsible for a general genetic instability and augmented chemoresistance [11]. Although these factors do not seem encouraging, ADEX cancers showed a better clinical outcome [11].

Finally, ADEX is an endocrine subtype which proved capable of impacting a patient’s hormonal balance: in fact, β-cell destruction is likely caused by the action of endocrine cells and PSCs in the TME [11]. PSCs are also responsible for a general genetic instability and augmented chemoresistance [11]. Although these factors do not seem encouraging, ADEX cancers showed a better clinical outcome [11].

Q9. Section 3.3.3. It should describe the role of arginase released by neutrophils. Authors can use: Sci Rep. 2021 Jun 15;11[1]:12574. doi: 10.1038/s41598-021-91947-0]

A9. We thank the reviewer to bring this reported research to our attention. Section 3.3.3 has been expanded citing the suggested article as below: “PDAC has demonstrated a high sensitivity to neutrophil activity, especially to lytic enzymes such as arginases-1 [10]. When incubated in an arginase-rich soil, PDAC cells showed augmented levels of Caspase-8 and Bap20 (indicating apoptosis), and cell death through an ER stress pathway was observed [10].

Q10. In section 5.4. I miss another figure explaining CAR-T in PDAC.

We followed the suggestion of reviewer, and we added the Figure 5 in section 5.4.4.

Q11. Some sentences must be review and the use of some words like "possible".

We followed, where possible, the constructive suggestion of the reviewer.

Round 2

Reviewer 2 Report

none

Reviewer 4 Report

Thanks so much for providing all my queries.

Some typos but well.